Association between coronary artery disease and incident cancer risk: a systematic review and meta-analysis of cohort studies

http://orcid.org/0000-0001-5785-8882 Chen Hsin-Hao 1 2 3
Lo Yi-Chi 4
Pan Wei-Sheng 4
http://orcid.org/0000-0002-6585-3894 Liu Shu-Jung 5
http://orcid.org/0000-0003-1894-0394 Yeh Tzu-Lin 1 6 5767@mmh.org.tw
Liu Lawrence Yu-Min 2 3 7 5954@mmh.org.tw
1 Department of Family Medicine, Hsinchu MacKay Memorial Hospital , Hsinchu City , Taiwan
2 MacKay Junior College of Medicine, Nursing, and Management , Taipei City , Taiwan
3 Department of Medicine, MacKay Medical College , New Taipei City , Taiwan
4 Department of Medical Education and Research, Hsinchu MacKay Memorial Hospital , Hsinchu City , Taiwan
5 Department of Medical Library, MacKay Memorial Hospital, Tamsui Branch , New Taipei City , Taiwan
6 Institute of Epidemiology and Preventive Medicine, College of Public Health, National Taiwan University , Taipei City , Taiwan
7 Division of Cardiology, Department of Internal Medicine, Hsinchu MacKay Memorial Hospital , Hsinchu City , Taiwan
Li Tian
Electronic publication date: 2023 Feb 23
Publication date: 2023
Volume: 11
Electronic Location ID: e14922
Received 2022 Oct 28; Accepted 2023 Jan 27
Copyright: © 2023 Chen et al.
Copyright year: 2023
Copyright holder: Chen et al.
License: This is an open access article distributed under the terms of the Creative Commons Attribution License, which permits unrestricted use, distribution, reproduction and adaptation in any medium and for any purpose provided that it is properly attributed. For attribution, the original author(s), title, publication source (PeerJ) and either DOI or URL of the article must be cited.
License URL: https://creativecommons.org/licenses/by/4.0/

Keywords: Cancer, Cohort, Coronary artery disease, Meta-analysis, Systematic review

Funding: The authors received no funding for this work.

==============================
Objective

Coronary artery disease (CAD) and cancer are the two leading causes of death worldwide. Evidence suggests the existence of shared mechanisms for these two diseases. We aimed to conduct a systematic review and meta-analysis to investigateassociation between CAD and incident cancer risk.

Methods

We searched Cochrane, PubMed, and Embase from inception until October 20, 2021, without language restrictions. Observational cohort studies were used to investigate the association between CAD and incident cancer risk. Using random-effects models, the odds ratio (OR) and 95% confidence interval (CI) were calculated. We utilized subgroup and sensitivity analyses to determine the potential sources of heterogeneity and explore the association between CAD and specific cancers. This study was conducted under a pre-established, registered protocol on PROSPERO (CRD42022302507).

Results

We initially examined 8,533 articles, and included 14 cohort studies in our review, 11 of which were eligible for meta-analysis. Patients with CAD had significantly higher odds of cancer risk than those without CAD (OR = 1.15, 95% CI = [1.08–1.22], I2 = 66%). Subgroup analysis revealed that the incident cancer risk was significantly higher in both sexes and patients with CAD with or without myocardial infarction. Sensitivity analysis revealed that the risk remained higher in patients with CAD even after >1 year of follow-up (OR = 1.23, 95% CI = [1.08–1.39], I2 = 76%). Regarding the specific outcome, the incident risk for colorectal and lung cancers was significantly higher (OR = 1.06, 95% CI = [1.03–1.10], I2 = 10%, and OR = 1.36, 95% CI = [1.15–1.60], I2 = 90%, respectively) and that for breast cancer was lower (OR = 0.86, 95% CI = [0.77–0.97], I2 = 57%) in patients with CAD than in those without CAD.

Conclusion

CAD may be associated with incident cancer risk, particularly for lung and colorectal cancers, in men and women as well as patients with or without myocardial infarction. Early detection of new-onset cancer and detailed cancer surveillance programs should be implemented in patients with CAD to reduce cancer-related morbidity and mortality.

Background

Cancers and coronary artery disease (CAD) are the two leading causes of death worldwide. They are closely associated with shared risk factors, which may indicate common biological characteristics, such as common pathways that result in smoking-related CAD and lung cancer (Das, Asher & Ghosh, 2019). Some studies have also suggested that cardiovascular diseases, such as myocardial infarction and cancer share similarities in terms of obesity, oxidative stress, and inflammation (Koene et al., 2016; Pischon & Nimptsch, 2016). People with mild CAD before cancer diagnosis may experience disease progression due to the cancer-induced proinflammatory and hypercoagulable states. Furthermore, CAD may cause a delay in the initiation of cancer treatment due to a decline in the patient’s heart condition or increased risk of surgery (Das, Asher & Ghosh, 2019). Thus, early detection of neoplasm in patients with CAD through appropriate strategies is critical for reducing future morbidity.

Some studies have reported increased incidence of CAD and stroke after cancer diagnosis. Various radio- and chemotherapeutic agents may affect the development and progression of cardiovascular disease (Arthurs et al., 2016; Navi et al., 2015; Zöller et al., 2012; Zhang et al., 2021). Further, several studies have indicated a high prevalence of occult cancer in patients with cardiovascular disease and reported that it is important to identify cancer risk factors as it may aid in developing new and effective preventive strategies (Corraini et al., 2018; Tybjerg, Skyhøj Olsen & Andersen, 2020; Wang et al., 2020).

In contrast, several recent clinical and epidemiological studies have revealed a link between myocardial infarction and new-onset cancer; (Dreyer & Olsen, 1998; Pehrsson, Linnersjö & Hammar, 2005) however, the findings were inconsistent and contradictory (Malmborg et al., 2018; Rinde et al., 2017). According to a systematic review, increased cancer risk after myocardial infarction was only significant in women and patients with certain cancers such as lung cancer. However, some of the review’s analytic findings were based on only two or three studies and it only included patients with myocardial infarction, not all patients with CAD (Li et al., 2019). Recently, a large cohort study demonstrated that atherosclerotic cardiovascular disease itself increased cancer incidence after a median follow-up of 1,020 days (Suzuki et al., 2017). Thus, the potential of CAD as a causal factor in cancer remains unknown. Furthermore, it has not yet been elucidated whether occult cancer occurs before the emergence of CAD. Therefore, this study aimed to conduct a comprehensive systematic review and meta-analysis to determine the association between CAD and incident cancer risk.

Methods

Data sources and study selection

This systematic review followed the Preferred Reporting Items for Systematic Reviews and Meta-Analyses (PRISMA) guidelines (Table S1) (Page et al., 2021). This protocol was registered into the PROSPERO International Prospective Register of Systematic Reviews (CRD42022302507).

The first author (Hsin-Hao Chen, HHC) and a medical librarian (Shu-Jung Liu, SJL) independently conducted an unrestricted search of electronic databases (Cochrane, PubMed, Embase (excluding Medline), and Taiwan Airiti Library) from inception until October 20, 2021. The following search terms were used: coronary artery disease, atherosclerosis, ischemic heart disease, myocardial infarction, neoplasms, cancer, and malignancy. The disagreements between the authors were resolved by a third reviewer (Tzu-Lin Yeh, TLY). We also examined potentially relevant studies in the references of relevant articles. Table S1 presents a complete description of the search strategies.

To identify eligible studies, we first removed duplicates. Two authors (Yi-Chi Lo, YCL and Wei-Sheng Pan, WSP) independently screened the titles and abstracts of each article, followed by a review of the full texts. If there was a disagreement, the third author (HHC) was consulted to reach consensus. Studies were included if they met the following criteria: (1) retrospective or prospective cohort studies; (2) studies investigating the association between fatal or nonfatal CAD and cancer risk; (3) studies wherein cancer occurred after CAD diagnosis; and (4) studies reporting adjusted cancer relative risk (RR), odds ratio (OR), and hazard ratio (HR) with 95% confidence interval (CI). Further, the exclusion criteria were as follows: (1) animal studies; (2) cross-sectional and case–control studies wherein cancer may have occurred before or concurrently with CAD; (3) nonobservational article types; (4) studies that did not report the relevant data for extraction; or (5) literature reviews, republished data, case reports, dissertations, editorial, letter, or conference abstracts. We initiated the formal screening of search results while registering the protocol into PRSOPERO because we were afraid that the COVID-19 pandemic would affect the writing and review process at that time.

Data extraction and quality assessment

Two authors (YCL and WSP) independently extracted the following data from each included article: first author, publication year, publication country, study design, CAD type, number of enrolled participants, age, follow-up duration, adjusted factors, cancer type, and main results (Table 1). Any disagreements were resolved through discussion with the third author (HHC). If any information was missing from the study results, the authors of original studies were contacted via email. The Newcastle Ottawa Scale (NOS) (Wells et al., 2022) was used by two authors (HHC and YCL) to independently assess the quality of the included studies. In cohort studies, the quality assessment tool (NOS) was used to rate each study in three domains—selection, comparability, and outcome—using a star system, with scores ranging from 0 to 9 stars (Zeng et al., 2015). The selection domain indicates representativeness of the exposed cohort, selection of the nonexposed cohort, and determination of exposure and outcome of interest that were absent at the beginning of the study. The comparability domain indicates whether exposed and nonexposed cohorts matched in the study design and/or whether confounders were adjusted for in the analysis. The outcome domain indicates whether the data were assessed accurately and whether the follow-up was adequate. If there was disagreement between two authors, the corresponding author (Tzu-Lin Yeh) made the final decision. A cohort study was considered to be of high quality if it received at least six stars.

Table 1 Characteristics of included studies.

Study	Country	CAD typea
and number of participants
(men %)	Age (years)	Follow up
(mean or median, years)	Adjusted factors	Cancer type	Main results
(CAD vs non-CAD or CAC = 0, presented as OR, HR, or RR with 95% CI)	
Dreyer & Olsen (1998)	Denmark	MI2
96,891 (67.97)	M: 63
F: 69	5.9 (1–17)	N/A	All	Total: 1.05 [1.03–1.07]
M: 1.03 [1.01–1.06]
F: 1.08 [1.04–1.12]	
Pehrsson, Linnersjö & Hammar (2005)	Sweden	MI2
N/A (65.20)	<80	9.3 (0–28)	Age	All	M: 1.08 [1.04–1.11]
F: 1.15 [1.09–1.21]	
Thomas et al. (2012)	USA	CAD1
547 (100)	66 (62–70)	4	Age, race, FH of prostate cancer, PSA, BMI, TRUS, HTN,DM, HL, aspirin, statin, alcohol, smoke, geographic area, DRE	Prostate	1.35 [1.08–1.67]	
Erichsen et al. (2013)	Denmark	MI2
297,523 (63.8)	69.4	3.1 (0–33)	Sex, age, duration	CRC	1.08 [1.05–1.11]	
Handy et al. (2016)	USA	CAD3
6,814 (47.1)	62.15 ± 10.2	10.2 (IQ: 9.7–10.7)	Age, sex, race, insurance, SES, BMI, PA, diet, smoke, drug,
SBP, DBP,
HTN drugs, TG, HDL, LL drugs, DM, aspirin	All	CAC >400 vs CAC = 0:
1.53 [1.18–1.99];
CAC = 0 vs CAC >0:
0.76 [0.63–0.92]	
Vinter et al. (2017)	Denmark	CAD3
28,549 (45.8)	49–66.5	M: 2.8
(IQ: 1.5–4.2);
F: 2.9
(IQ: 1.7–4.3)	Age, BMI, DM, smoke, LL drugs, HTN drugs, Cr, HF.	All	M:
CAC = 1–99:1.07 [0.83–1.39]
= 100–399:1.24 [0.94–1.63]
= 400–999:0.88 [0.62–1.25]
≥1,000:0.96 [0.66–1.41]
F:
CAC = 1–99:0.96 [0.77–1.19]
= 100–399:0.99 [0.75–1.31]
= 400–999:1.11 [0.76–1.62]
≥1,000:1.16 [0.73–1.83]	
Rinde et al. (2017)	Norway	MI2
1,747 (62)	62	15.7	Age, sex, BMI, SBP, DM, HDL, smoke, PA, Edu.	All	All: 1.46 [1.21–1.77]
M: 1.29 [1.02–1.62]
F: 1.65 [1.19–2.29]	
Suzuki et al. (2017)	Japan	CAD1
32095 (59)	65 ± 16	2.8 (IQ: 1.8–3.7)	Age, sex, lifestyle-related disease, smoke, f/u periods	All	1.42 [1.02–1.96]	
Berton et al. (2018)	Italy	CAD1
589 (70)	67 (58–74)	17	N/A	All	Incidence: 17.8 per 1,000 person-years	
Malmborg et al. (2018)	Denmark	MI2
122,275 (61.2)	M: 59.2
(49.5–69.5)
F: 68.5
(58.1–76.0)	0.5–17	Age, sex, calendar year, HTN, HL, DM, COPD, SES	All	Total: 0.97 [0.92–1.01]
M: 0.97 [0.91–1.03]
F: 0.99 [0.92–1.06]
(exclude first 6 months)	
Kwak et al. (2020)	Korea	CAD2
753,678 (72.1)	63.5	4.56 (IQ:3.06–6.13)	Age, sex, income, DM, BMI, smoke, alcohol, PA.	All	1.06 [1.04–1.09];
exclude first year:
1.02 [0.99–1.05]	
Mirbolouk et al. (2020)	USA	CAD3
Nonsmoker:
48,331 (65.5)	54.6 ± 10.6y	11.9 (IQ:10.2–13.3)	Age, sex, HL, FH of CAD, HTN, DM.	All	CAC = 1–99: 1.05 [0.84–1.30]
= 100–399:1.19 [0.93–1.51]
>400: 1.19 [0.92–1.55]	
Smoker
5,147 (67.6)	52.8 ± 9.9	11.9 (IQ: 10.2–13.3)	Age, sex, HL, FH of CAD, HTN, DM.	All	CAC = 1–99: 0.83 [0.48–1.43]
= 100–399:1.06 [0.60–1.89]
>400:1.85 [1.07–3.22]	
Peng et al. (2020)	USA	CAD3
66,636 (67%)	54.4 ± 9.6	12.3 ± 3.9	Age, sex, HTN, HL, smoke, DM, FH of CAD.	All	CAC = 1–399:1.10 [0.95–1.28]
= 400–999:1.18 [0.94–1.47]
>1,000:1.51 [1.19–1.91]	
Dzaye et al. (2021)	USA	CAD3
6,271 (47.3%)	61.7 ± 10.2	12.9 ± 3.1	Age, sex, ethnicity, BMI, PA, SES, Edu, insurance, smoke, diet	Lung/rectal	CAC = 1–99: 1.2 [0.81–1.78]
= 100–399:1.87 [1.20–2.92]
>400:2.01 [1.20–3.35]	
Prostate	CAC = 1–99:1.52 [1.00–2.30]
= 100–399:1.07 [0.62–1.85]
≥400:1.13 [0.65–1.95]	
Breast/uterine/ovary	CAC = 1–99 = 0.76 [0.44–1.30]
= 100–399 = 0.54 [0.24–1.19]
≥400 = 1.13 [0.51–2.51]	
Notes:

a CAD diagnosed by (1) hospital medical records (2) discharge records with Internal Classification of Disease; (3) computed tomography scan.

BMI, Body mass index; CAC, Coronary Artery Calcium Score; CAD, Coronary artery disease; Cr, Creatinine; CRC, colorectal cancer; DBP, diastolic blood pressure; DM, Diabetes mellitus, Drug medication, DRE, Digital rectal examination; Edu, education; F, female; FH, Family history; F/U, follow up; HL, Hyperlipidemia; HF, Heart failure; HTN, Hypertension; IQ, interquartile range; LDL, high-density lipoprotein cholesterol; LL drugs, lipid-lowering drugs; M, male; N/A, Not available; PA, physical activity; SBP, Systolic blood pressure; SES, socioeconomic status; RETRO, retrospective; SIR, standardized incidence ratios; TG, total cholesterol; TRUS, transrectal ultrasound volume; USA, United States of America.

Statistical analysis and data synthesis

We calculated pooled ORs with 95% CIs to estimate incident cancer risk in patients with CAD and compared it with that in patients without CAD. For our meta-analysis, we used statistical computing software R, version 4.1.2 (R Core Team, 2021), primarily the Comprehensive R Archive Network package “metagen” (Software, 2022). Subsequently, we employed a random-effects model based on the DerSimonian and Laird’s method with an assumption of nonidentical true effect sizes (DerSimonian & Laird, 1986). These results were presented as forest plots. Furthermore, heterogeneity among studies was quantified using Cochran’s Q test and I2 statistics, and a p-value of <0.05 in the Q test or I2 value of >50% indicated the presence of heterogeneity (Higgins & Thompson, 2002). Subgroup analysis was determine to assess the potential origins of heterogeneity. We did not perform a meta-regression analysis using patient characteristics, as some studies did not provide enough study-level variable information (Pehrsson, Linnersjö & Hammar, 2005; Dzaye et al., 2021). Thus, this method would have been unsuitable, according to the methodological standards for meta-analysis and qualitative systematic reviews (Rao et al., 2017). We investigated the association between CAD and different cancers, including lung, colorectal, breast, liver, and prostate cancers. To assess the robustness of the results, we performed a sensitivity analysis that included only studies with a follow-up time of >1 year. The risk of publication bias was assessed using funnel plots and Egger’s test (Egger et al., 1997).

Results

Study characteristics and quality assessment

Figure 1 presents the article selection flowchart. Initially, we obtained 8,533 articles from databases and by hand searching. Subsequently, we removed duplicates, reviewed titles and abstracts, and retrieved and evaluated 25 full-text articles for eligibility. After excluding articles with duplicate populations or those incompatible with the inclusion criteria, our systematic review included 14 cohort studies, 11 of which were eligible for meta-analysis (Fig. 1).

Figure 1 Flow diagram for selection of articles.

Table 1 summarizes the general demographic characteristics of the included studies in the systematic review. Of the included studies, only two (Suzuki et al., 2017; Kwak et al., 2020) were conducted in Asia, whereas other studies were from USA or Europe. Four studies included patients with myocardial infarction identified via discharge diagnosis with Internal Classification of Disease (ICD) codes, (Pehrsson, Linnersjö & Hammar, 2005; Malmborg et al., 2018; Rinde et al., 2017; Erichsen et al., 2013) whereas other studies included patients with CAD identified via hospital medical records, discharge diagnosis with ICD codes, or computed tomography scan with coronary artery calcium (CAC) score of >0. The duration of follow-up ranged from <1 year to a maximum of 33 years. Furthermore, we confirmed that the diagnosis of CAD was made before the occurrence of cancer in all included studies. Considering the cancer type, most studies investigated the incidence of all cancers, whereas other studies only assessed specific cancers, such as colorectal cancer, (Dzaye et al., 2021; Erichsen et al., 2013) or cancers specific to men (prostate) or women (Dzaye et al., 2021; Thomas et al., 2012). Regarding the outcomes, a study only reported the incidence rate, (Berton et al., 2018) whereas other studies provided the overall or subgroup effect estimates of RR, OR, and HR with 95% CI.

In our study quality assessment, we observed that only one study did not report the items of selection and comparability domain and, as such, did not meet our criteria (Dreyer & Olsen, 1998). All other included studies received at least six of nine stars on the NOS quality assessment scale, indicating high quality. Tables S3 presents the detailed results.

Results of meta-analysis

We pooled 11 studies for meta-analysis, which included >1,321,978 patients; however, one of these studies (Pehrsson, Linnersjö & Hammar, 2005) did not specify the number of participants. Patients with CAD had significantly higher odds of cancer risk than those without CAD (OR = 1.15, 95% CI = [1.08–1.22], I2 = 66%; forest plot shown in Fig. 2). Subgroup analyses were performed based on the heterogeneity in the country and CAD type of patients. Patients with CAD had significantly higher odds of cancer risk than those without CAD in non-Asian regions (OR = 1.15, 95% CI = [1.08–1.23], I2 = 67%; Fig. S1). Furthermore, Asian patients with CAD showed nonsignificantly higher odds of cancer risk than those without CAD (OR = 1.17, 95% CI = [0.89–1.53], I2 = 67%; Fig. S1). We also conducted a subgroup analysis by CAD subtype, which revealed that those with or without myocardial infarction had significantly higher odds of cancer risk among patients with CAD than among those without CAD (OR = 1.11, 95% CI = [1.00–1.23], I2 = 89% and OR = 1.17, 95% CI = [1.08–1.27], I2 = 51%, respectively; Fig. S2).

Figure 2 Forest plot of incident cancer risk, comparing participants with CAD as those without CAD.

CAD, coronary artery disease; CI, confidence interval; OR, odds ratio; se, standard error; TE, treatment effect.

Subgroup analysis by sex

We also performed pooled analyses in a random-effects model based on sex. This analysis was conducted when the studies indicated the odds of cancer risk by individual sex. After pooling seven studies, (Dreyer & Olsen, 1998; Pehrsson, Linnersjö & Hammar, 2005; Malmborg et al., 2018; Rinde et al., 2017; Dzaye et al., 2021; Thomas et al., 2012; Vinter et al., 2017) the overall risk of cancer incidence in men with CAD was higher than that in those without CAD (OR = 1.12, 95% CI = [1.03–1.22], I2 = 61%; Fig. 3(1)). Furthermore, after pooling six studies, (Dreyer & Olsen, 1998; Pehrsson, Linnersjö & Hammar, 2005; Malmborg et al., 2018; Rinde et al., 2017; Dzaye et al., 2021; Vinter et al., 2017) women with CAD showed a higher incident cancer risk than those without CAD (OR = 1.08, 95% CI = [1.00–1.16], I2 = 56%, Fig. 3(2)).

Figure 3 Forest plot of incident cancer risk, comparing participants with CAD as those without CAD by individual gender.

(1) Men (2) Women. CAD, coronary artery disease; CI, confidence interval; OR, odds ratio; se, standard error; TE, treatment effect.

Subgroup analysis by different outcome

We determined whether CAD exerted different effects on different types of cancer. Patients with CAD had a significantly higher risk of colorectal and lung cancers than those without CAD (OR = 1.06, 95% CI = [1.03–1.10], I2 = 10%; Fig. 4(1)) and (OR = 1.36, 95% CI = [1.15–1.60], I2 = 90%, respectively; Fig. 4(2)), as determined after pooling four (Pehrsson, Linnersjö & Hammar, 2005; Kwak et al., 2020; Erichsen et al., 2013; Vinter et al., 2017) and five (Dreyer & Olsen, 1998; Pehrsson, Linnersjö & Hammar, 2005; Malmborg et al., 2018; Kwak et al., 2020; Vinter et al., 2017) studies, respectively. However, according to the odds of breast cancer risk in five studies, (Dreyer & Olsen, 1998; Pehrsson, Linnersjö & Hammar, 2005; Malmborg et al., 2018; Kwak et al., 2020; Vinter et al., 2017) a lower risk was observed among patients with CAD than among those without CAD (OR = 0.86, 95% CI = [0.77–0.97], I2 = 57%; Fig. 4(3)). Furthermore, compared with patients without CAD, a nonsignificantly increased risk of prostate and liver cancers was observed in those with CAD (OR = 1.04, 95% CI = [0.94–1.16], I2 = 72%; Fig. S3(1) and OR = 1.03, 95% CI = [0.88–1.21], I2 = 59%, respectively; Fig. S3(2)), as determined after pooling seven (Dreyer & Olsen, 1998; Pehrsson, Linnersjö & Hammar, 2005; Malmborg et al., 2018; Dzaye et al., 2021; Kwak et al., 2020; Thomas et al., 2012; Vinter et al., 2017) and three (Dreyer & Olsen, 1998; Pehrsson, Linnersjö & Hammar, 2005; Kwak et al., 2020) studies, respectively.

Figure 4 Forest plot of incident cancer risk, comparing participants with CAD as those without CAD by individual cancer type.

(1) Colorectal cancer (2) Lung cancer (3) Breast cancer. CAD, coronary artery disease; CI, confidence interval; OR, odds ratio; se, standard error; TE, treatment effect.

Sensitivity analysis and publication bias

We analyzed six studies in which all patients had a follow-up time of >1 year (Rinde et al., 2017; Dzaye et al., 2021; Kwak et al., 2020; Erichsen et al., 2013; Handy et al., 2016; Peng et al., 2020). The incident cancer risk was still higher in patients with CAD than in those without CAD (OR = 1.23, 95% CI = [1.08–1.39], I2 = 76%; Fig. S4). Funnel plots revealed asymmetry for publication bias, as shown in Fig. S5. In addition, Egger’s test revealed a significant publication bias (p = 0.06).

Discussion

Our meta-analysis revealed that patients with CAD had significantly higher odds of cancer risk than those without CAD among cohort studies. Subgroup analysis indicated that cancer risk was significantly higher in both men and women, those with and without myocardial infarction, and non-Asian patients. Moreover, for specific cancer types, patients with CAD had a higher risk of colorectal and lung cancers, nonsignificantly higher risk of prostate and liver cancers, and lower risk of breast cancer.

A previous systematic review of myocardial infarction based on only three studies revealed that the incident cancer risk in the test group was nonsignificantly higher (OR = 1.08, 95% CI = [0.97–1.19]) than that in the control group. However, subgroup analysis revealed that the overall cancer risk was higher in women and during the first 6 months following myocardial infarction diagnosis (Li et al., 2019). Further, our meta-analysis of eleven studies revealed a significantly higher incident cancer risk in patients with CAD with or without myocardial infarction. One of the differences in the outcomes of patients with myocardial infarction is the number of cohort participants included in the meta-analysis. As the 1998 study by Dreyer & Olsen (1998) in Denmark comprised only a small proportion (96,891 people) of the 2013 study by Erichsen (297,523 people), (Erichsen et al., 2013) we included a large cohort instead of a small cohort. Further, our meta-analysis evaluated patients without myocardial infarction via CAC, percutaneous coronary intervention (PCI), or hospital discharge records to comprehensively assess cancer risk in patients with CAD.

CAD and incident cancer risk are mainly associated because of the presence of shared risk factors. As summarized in the study by Hasin et al. (2016) cancer may be caused by treatment modalities or biological changes related to cardiovascular diseases. Other reviews have also indicated that inflammatory cytokines, such as interleukin(IL)-1, IL-6, IL-10, tumor necrosis factor-α, macrophage migration inhibitory factor, and transforming growth factor-β, are involved in tumor initiation and progression (Amin et al., 2020; Leiva et al., 2021). In addition to inflammation during the development of atherosclerosis and cancer, a recent review revealed that age-related mutations, obesity, smoking, and diabetes are overlapping risk factors between cancer and CAD (Leiva et al., 2021). Additionally, some observational studies have reported that noncardiac causes, such as malignancies, are responsible for most later deaths in patients with myocardial infarction treated with PCI (Pedersen et al., 2014; Spoon et al., 2014).

Conversely, some studies have suggested that the increased cancer risk immediately after myocardial infarction can be attributed to other confounding factors, such as surveillance bias, rather than myocardial infarction itself. Patients with myocardial infarction had frequent clinical appointments and underwent more diagnostic examinations, especially in the first few months after the event, which may increase the likelihood of early cancer detection (Malmborg et al., 2018; Li et al., 2019). This situation is not only observed in patients with myocardial infarction but also in those without. Other studies have shown that occult cancers could have occurred before the cardiovascular event if cancer incidence is observed immediately after the start of myocardial infarction follow-up (Hasin, Iakobishvili & Weisz, 2017). In some patients, an underlying malignancy can cause an ischemic stroke. The effects of the coagulation cascade, tumor mucin secretion, infections, and nonbacterial endocarditis may contribute to the mechanisms (Selvik et al., 2015). Thus, occult cancer may also contribute to the development of CAD. However, our sensitivity analysis revealed that patients with CAD continue to have an increased incident cancer risk after >1 year of follow-up, which differs from the meta-analysis based on only two studies reporting that cancer risk is only significant in the first 6 months. Another study revealed that although the cancer risk is the highest in the first year following myocardial infarction, cancer develops over time (Malmborg et al., 2018). According to a recent large-scale cohort study, atherosclerotic cardiovascular disease increases the incident cancer risk after a median follow-up of 1,020 days (Suzuki et al., 2017). Moreover, the risk is increased when patients with CAD concomitantly have aortic and peripheral artery disease with a median follow-up of 3 years (Suzuki et al., 2022). Therefore, CAD may affect long-term cancer incidence.

Our study revealed that CAD events increased the risk of lung and colorectal cancers but decreased the risk of breast cancer. We determined that “smoking,” a well-known cause of lung and colorectal cancers, was a common risk factor. This may account for some of our findings that indicate that the risk of both cancers was significantly increased after CAD (Dekker et al., 2019). Another reason for an increase in lung cancer incidence may be that cardiac scanning includes the lungs; thus, lung cancers account for most detected cancers (Vinter et al., 2017). Diabetes is a classic risk factor for CAD and is also related to elevated risk of cancer, especially colorectal cancer (Leiva et al., 2021). A study showed that patients with diabetes had a 20–38% higher cancer risk than those without diabetes (Yuhara et al., 2011). Moreover, modifiable environmental risk factors, such as obesity, lack of physical activity, and Westernized diet, may predispose individuals to CAD and colorectal cancer (Keum & Giovannucci, 2019). According to two large prospective cohort studies, a high intake of animal fat or processed red meat and low intake of fiber could increase the risk of CAD and colon cancer (Al-Shaar et al., 2020; Willett et al., 1990). One possible explanation for the lower risk of breast cancer in our study is life-long aspirin treatment, as recommended by CAD guidelines, (Qiao et al., 2018) which may also affect carcinogenesis. Large-scale cohort studies have consistently demonstrated the protective effects of low-dose aspirin for treating breast cancers (Qiao et al., 2018; Yang et al., 2017). However, there is limited evidence to support the association between CAD and breast cancer and we cannot exclude the possible selection bias; therefore, more research is warranted in this regard.

This is the first study to conduct a comprehensive review and meta-analysis of the association between CAD and incident cancer risk with regard to patients with or without myocardial infarction as well as different cancer types. However, there are some limitations that must be addressed. First, our meta-analysis had significant publication bias, indicating that some nonsignificant studies are not published. This would weaken the positive association between CAD and incident cancer risk observed in our study. However, current evidence was the best available, and all studies, including several population-based cohort studies, were of moderate-to-high quality. Second, not all included studies could distinguish the length of follow-up and different cancer types. Our findings showed that the cancer risk remains elevated even at 1 year of follow-up after a CAD event, which contradicts the findings of the previous two studies (Malmborg et al., 2018; Kwak et al., 2020). According to our subgroup analysis, CAD may have different effects on different cancer types. Additional studies with subgroup analysis of follow-up time and different types of cancer are thus warranted to investigate the association between CAD and incident cancer risk. Third, most studies did not provide data regarding heart failure or left ventricular ejection fraction. Recently, Meijers et al. (2018) indicated that heart failure stimulates tumor growth via cardiac-excreted circulating factors. Furthermore, heart failure is associated with cancer incidence (Hasin et al., 2016) and could become a confounding factor in future research.

Conclusions

Our analysis of newly published data suggested an increased risk of incident cancer after a CAD event, particularly for lung and colorectal cancers. This increased risk was observed in men and women with or without myocardial infarction. Although this trend may be attributable to several common risk factors and underlying pathophysiologic mechanisms such as inflammation, patients with a history of CAD are still more likely to develop cancer. As CAD and cancer are the two leading causes of death, treatment of any one disease may affect the occurrence of the other. Therefore, more research is warranted regarding the causes of malignancy. Further, detailed cancer surveillance and possible interventions in the CAD population should be implemented to reduce cancer-related morbidity and mortality.

Supplemental Information

Supplemental Information 1 PRISMA checklist.

Click here for additional data file.

Supplemental Information 2 Raw data.

Click here for additional data file.

Supplemental Information 3 Contribution and rationale.

Click here for additional data file.

Supplemental Information 4 Supplementary tables and figures.

Click here for additional data file.

Supplemental Information 5 Replicate check.

Click here for additional data file.

We would like to thank the MacKay Memorial Hospital librarian, Pei-jin Li, for examining the references. We would like to thank Romy E. for editing and proofreading the manuscript.

Additional Information and Declarations

Competing Interests

Author Contributions

Data Availability

The authors declare that they have no competing interests.

Hsin-Hao Chen conceived and designed the experiments, performed the experiments, analyzed the data, prepared figures and/or tables, authored or reviewed drafts of the article, and approved the final draft.

Yi-Chi Lo performed the experiments, analyzed the data, prepared figures and/or tables, and approved the final draft.

Wei-Sheng Pan performed the experiments, analyzed the data, prepared figures and/or tables, and approved the final draft.

Shu-Jung Liu performed the experiments, prepared figures and/or tables, and approved the final draft.

Tzu-Lin Yeh conceived and designed the experiments, authored or reviewed drafts of the article, and approved the final draft.

Lawrence Yu-Min Liu conceived and designed the experiments, authored or reviewed drafts of the article, and approved the final draft.

The following information was supplied regarding data availability:

The raw data are available in the Supplemental File.

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
