# Peer review of "Association between coronary artery disease and incident cancer risk: a systematic review and meta-analysis of cohort studies"

_PeerJ, doi:10.7717/peerj.14922_

## Round 0.1 · original submission · Major Revisions

A detailed point to point response letter (editor + reviewers), manuscript with tracked font are necessary for further process when they submit their revised version of paper. Please upload the response letter as a PDF file in the Supplementary materials in the online submission system. Please give detailed response. Please write down what changes have been made in the response letter, rather than reading the manuscript to find what has been revised. And please note that any uncompleted or improper corrections by the authors during this revision may lead to rejection.

1. The language needs revision by a proficient English speaker, accompanied with a certificate of language editing service and a manuscript with tracked editing records as a Supplemental File
2. There are some spelling mistakes and format errors, e.g. the typesetting is chaotic.
3. Similar studies were found in PubMed. Authors shall cite part of them and discuss what is new and different from this article.
4. Provide supporting references for Section Methods
5. Please provide a duplicate check report by authors as a supplementary file (total < 20%, each < 2%).
6. Add China for author affiliations in the title page.

Reviewer 1 ·

Basic reporting

no comment

Experimental design

- Please write the author's name first, followed by an abbreviation. (line 92)
- Please add a reference to Newcastle Ottawa Scale (NOS) and a brief description of the star rating system. (line 120-121)
- Please specify the exact version of R. (e.g. R 4.x.x) RStudio is an integrated development environment (IDE) for R. (line 129-130)

Validity of the findings

Major comments
1. As the authors point out, the reason cancers are often diagnosed after CVD events is because they share the same risk factors. Therefore, the possibility of cancer diagnosis may increase immediately after a CVD event as well as thereafter. This study provides reasons for clinicians to take a closer look at the test results and symptoms of patients who have suffered a CVD event. If the authors mention this in more detail in the discussion, it will be able to deliver a clearer message to researchers and clinicians.

2. As the authors noted, this study cannot establish a causal relationship between CVD and cancer. However, the concluding part of the abstract gives the impression that there is causality. (line 54-56) Perhaps 'association' or 'relationship' would be more appropriate than 'risk factor' in this case.

3. A particularly interesting aspect of this study is that the association between colorectal cancer and CVD was higher than that of other cancers. As noted by the authors, a Western diet may be a common risk factor. (line 259) As a more specific cause, excessive intake of processed meat or low dietary fiber intake can be discussed. I recommend adding this to the discussion.

Additional comments

The authors found an association between CVD and cancer in a well-described meta-analysis. This meta-analysis showed that the likelihood of cancer diagnosis increases in both men and women after fatal or non-fatal CVD. I mostly agree with the authors' research methodology and conclusions.

Reviewer 2 ·

Basic reporting

no comment

Experimental design

The author mentioned that he retrieved 8,533 articles from databases by hand searching and then filtered them to be 24 articles, is my understanding correct? if yes how do I search about 8500 articles manually, if 24 articles are sufficient why you didn't make this criterion at the beginning so that I would save considerable time?

Validity of the findings

The author makes a very important observation, he investigated a link between CAD and different cancer types, including lung, colorectal, breast, liver, and prostate cancers

Additional comments

In general, this is exciting research in the medical that shows a causal relation between disease or the disease and the organ, the writing way is some good and the language is clear, but I prefer in the future work to add a background section to define some terminology, in addition, the methodology and experiment needs to add some figures to summarize the observation.

Reviewer 3 ·

Basic reporting

.

Experimental design

.

Validity of the findings

.

Additional comments

This study attempted a systemic review and meta-analysis of cohort studies regarding a clinical question whether there is an association between coronary artery disease and subsequent cancer developments. The investigators picked up a total of 14 cohort studies from the database of Cochrane, PubMed, and Embase until October, 20, 2021. They analyzed heterogeneity of those articles using a random-effects model by DerSimonian and Laird’s method and also Cochran’s Q test and I2 statistics showing a quite high variations between those studies. Nevertheless, they concluded that patients with coronary artery disease had significantly higher risks than those without coronary artery disease. The present study, if corrected from the viewpoints of meta-analysis, may recognize the readers a crucial importance of the concept of cardio-oncology as reported by Suzuki et al. (ref. no. 16) that a presence of atherosclerotic cardiovascular disease may directly and/or indirectly have a potential risk for cancer development. However, to overcome the critical limitations of the present study, the authors need to clear the major issues mentioned below.

First, the authors need to reconfirm the credibility of the present selected articles. As shown in Table 1, a total of 14 studies were selected for the present study, whereas Dreyer’s report (ref. no.11) subjected one-year survivors after acute myocardial infarction without detailed clinical information but not any other coronary artery diseases, Thomas II’s report (ref. no. 27) analyzed only 547 men with coronary artery disease which revealed a number of 6,390 patients in the present Table 1, a number of 32,095 patients in Suzuki’s report (ref. no. 16) included not only coronary artery disease but other atherosclerotic cardiovascular diseases, Berton’s report (ref. no. 28) and Malmborg’s report (ref. no. 13) subjected only acute coronary syndromes, Kwa’s report (ref. no.25) analyzed only patients underwent coronary angioplasty without any clinical information. Furthermore, a couple of clinical studies to assess the relation between a degree of coronary artery calcium and cancer incidence was included in this meta-analysis (ref. no. 22, 29-31). Thus, the authors may need a lot of attention to avoid a considerable confusion between coronary atherosclerosis and coronary artery disease for the present study design. Anyway, a great deal of clinical variations as mentioned above may produce a crucial loss of confidence for the present results.

Second, there was no reference as to the article by Mirboulouk et al. described in Table 1. Also, Dzaye et al.’s report (ref. no.22) was published in 2022 but not 2021, which means the end of timeline to pick up the articles is not until October, 20, 2021.

Finally, to encourage the authors to re-establish the present meta-analysis, the reviewer welcomed to provide a scientific information as to a recent study by Suzuki el al. entitled “Polyvascular disease and the incidence of cancer in patients with coronary artery disease” published in JMA J. 2022;5(4):498-509 (DOI:10.31662/jmaj.2022-0098 https://www.jmaj.jp/) .

---

## Round 0.2 · accepted · Accept

The authors have addressed my questions.

Reviewer 1 ·

Basic reporting

no comment

Experimental design

no comment

Validity of the findings

no comment

Additional comments

I appreciate the detailed responses from the authors. There are no further comments.

Reviewer 2 ·

Basic reporting

The paper is well written, and the related study, background, figures, and methodology are sufficient and clear.

I see some words stick together on page 4 like- investigateassociation, withoutlanguage, please fix them.

Experimental design

No comments

Validity of the findings

No comments

Additional comments

This research type is essential and exciting, it finds an association between CAD and incident cancer risk
The authors depend on various databases: Cochrane, PubMed, and Embase Without language restrictions.
This study aimed to conduct a comprehensive systematic review and meta-analysis to determine the association between CAD and incident cancer risk.

I highly recommend this type of study that creates awareness among people about the relationship between diseases to each other and also helps doctors pay attention to the causes of disease, which increases the chance of treating the disease from an early stage
From a technical point of view, as a specialist in computer science and machine learning, I recommend using this type of technology, which will undoubtedly increase the quality of this type of research.